# Study on Safety Coefficient of Sedimentary Bauxite Strip Pillar under Valley Terrain

**DOI:** 10.3390/ijerph191710991

**Published:** 2022-09-02

**Authors:** Lichun Jiang, Huazhe Jiao, Bo Xie, Han Yang

**Affiliations:** 1School of Civil Engineering and Transportation, South China University of Technology, Guangzhou 510640, China; 2Institute of Safety Science and Engineering, South China University of Technology, Guangzhou 510640, China; 3State Collaborative Innovation Center of Coal Work Safety and Clean-Efficiency Utilization, Jiaozuo 454003, China; 4School of Civil Engineering, Henan Polytechnic University, Jiaozuo 454003, China

**Keywords:** valley terrain, strip pillars, safety coefficient, stress concentration coefficient, sedimentary bauxite

## Abstract

The underground sedimentary bauxite ore body in Shanxi province has a shallow burial depth; the valley terrain caused stress concentration on a pillar which affected the pillar’s safety and goaf stability. This paper proposed a pillar safety coefficient calculation method affected by the goaf structural parameters and the valley terrain, which was based on a pillar mechanics analysis under the valley terrain. The results show that the overlying valley terrain will cause stress concentration on the pillar, reducing the adequate bearing capacity and the pillar stability. The increase of the goaf span b and the height of the pillar h is extensively detrimental to pillar stability. Meanwhile, increasing the pillar burial depth would cause the pillar to weaken, but can effectively decrease the influence of the valley terrain. Furthermore, when the angle between the goaf strike and the valley strike *β* < 50°, *β* has a more significant impact on the stress concentration and safety coefficient. The stability of an underground sedimentary bauxite pillar is calculated by the method, the result complied with the actual situation.

## 1. Introduction

Sedimentary bauxite ore bodies in Shanxi Province, China generally have shallow burial depths, and the overlying rock and soil bodies are mainly loess with a thickness of tens to hundreds of meters, it is usually mined by the strip room-pillar method. Under rain erosion, the sandy soil type loess is seriously lost, and the surface often forms a unique valley terrain [1,2,3,4]. Mine production practices show that many problems such as stress concentration and shear-slip damage often occur in pillars under the valley terrain, which seriously threatens the safety and stability of the goaf [5,6,7,8,9]. Therefore, it is of great practical significance to grasp the influence of valley terrain on the stability of the underlying sedimentary bauxite strip pillars to ensure the safe mining of such mines.

At present, domestic and foreign scholars have been done many scientific research studies on issues related to pillar stability, and have achieved many research results. González-Nicieza et al. [10], based on the Bieniawski rock mass rating, considered the characteristics of the overlying rock mass, and gave a calculation method for the safety coefficient of the pillar in the case of fracturing failure and shear instability. Esterhuizen et al. [11] proposed a design method for the calculation of pillar power and the selection of a safety coefficient by considering the pillars defects and the strength of intact rock blocks through the investigation of the stable pillar and failure pillar of a large number of sedimentary quarries in the eastern and midwestern United States. Zhao et al. [12] analyzed the pillar bearing mechanism using Platts theory, deduced the critical dimension formula of pillar failure, and studied the pillar stability of certain phosphate rock by numerical method. Zhao et al. [13] constructed the pillar energy limit state equation based on the principle of energy conservation and catastrophe theory, deduced the critical stability criterion of the pillar, and studied the change of the pillar stability under different external loads. Luo et al. [14] used the theory of elasticity and Mohr-Coulomb criterion to construct an analytical formula for the safety coefficient of the inclined pillar. They studied the influence of the ore body dip angle and lateral pressure coefficient on the stability of the inclined pillar. To sum up, most of the existing pillar stability studies are based on the fact that the surface is horizontal, and the safety coefficient is only calculated from the strength of the pillar itself, ignoring the influence of the surface topography.

Shallowly buried surface ore bodies are prone to generating uneven loads due to the influence of topography, which is seriously detrimental to the safety and stability of coal mining. When mining shallowly buried and thick coal seams, the in-situ stress field of the coal seam is affected by the gully terrain, and the shear stress concentrates on the surface to cause geological hazards such as landslides or gully collapse [15]. Li et al. [16] proposed a new method to study the distribution of stress field, displacement field and damage field of overlying strata in coal mining under valley terrain, aiming at the problem of geological damage that easily occurs when mining shallow coal seams in gully areas.

Based on the control of subsurface and surface damage of shallow thick coal seam mining in the gully region, the slicing mining and filling mining of a thick coal seam is put forward. Wang et al. [17] systematically investigated the overburden movement and strata behavior at a fully mechanized top coal caving (FMTCC) face in a thick coal seam in a loess-covered gullied region by field measurement and theoretical analysis. A comparative analysis was performed to examine how the attitude of surface gullies and the structure of overburden affect face support resistance. The study on influence laws of strata behaviors is the fundamental guarantee of safety mining for shallow coal seams beneath gully terrain. The theoretical analysis shows that the rotation and breaking of essential stratum beneath gully bottom under nonuniform load is the fundamental cause of vital dynamic strata behaviors [18].

The uneven load brought by the surface topography puts higher requirements for the safety and stability of the pillar. Kumar et al. [19] proposed a probabilistic method to analyze the stability of coal pillars, providing additional design criteria on the traditional design criteria for pillar stability safety coefficient to select the optimal safety coefficient of pillars to improve the recovery ratio of coal mine. Jiang et al. [20] proposed a method to optimize customized short-wall mining parameters in deep and extra-thick coal seams, aiming to meet the coal pillar stability requirements and achieve the maximum coal recovery rate. Guo et al. [21] found that the width of the plastic zone of coal pillars in deep mines increased significantly through field measurements and three-dimensional similar material simulation experiments. Wagner et al. [22] evaluated the effect of panel size on pillar loads through numerical simulation, and the results showed that pillar loads in deep mines were strongly influenced by the lateral and vertical extent of the mining panels. Zhu et al. [23] established a FLAC3D intermittent mining model according to the characteristics of the goaf and the particularity of the internal load-bearing structure, evaluated the stability of the pillar, and revealed the law of long-term stability and distribution of the overlying load. Sherizadeh et al. [24] used 3DEC numerical simulation to evaluate the influence of pillar size on the stability of the top and found that geological conditions and mining coefficient s are important factors affecting the stability of the top. To maintain the stable state of the stope for a long time, the cross-sectional size of the pillar itself must also meet the strength requirements during design [25,26,27], to avoid the height of the pillar being too small to bear the overlying load, resulting in the pillar being crushed.

Currently, only numerical simulation is used for the regional stability calculation of shallow buried ore bodies [28,29]. Although it can meet the theoretical calculation requirements to a certain extent, it seriously neglects the influence of the goaf strike, the structural parameters, and the surface topography on the stability of the pillar. Based on the field investigation of a sedimentary bauxite mine in Shanxi, this paper analyzes the stress characteristics of the strip pillar under the valley terrain. It builds a mechanical model of the pillar under the valley terrain. Based on the Hoek–Brown strength criterion [30], the calculation method of pillar safety coefficient under valley terrain is proposed, and the influence of three types of parameters on the stability of the pillar, such as the structural parameters of the goaf, the valley terrain parameters and the goaf strike are analyzed. Finally, the reliability of the calculated conclusions is verified by using the recorded data of the project site.

## 2. Mechanical Model of Pillar under Valley Terrain

Taking the underlying sedimentary bauxite mining area of a typical valley terrain in Shanxi as an example to study (Figure 1), the distribution of the goaf is shown in Figure 2. The width of the pillar is *a*, the span of the goaf is *b*, the thickness of the overlying rock and soil layer in the goaf is *H*, the pillar load is *P*, the valley slope angle of the surface is α.

As the mining face moves forward, the pillar in the goaf will alternately pass through the valley terrain region and the flat terrain area. Different regions have different loads on the pillars. In the peaceful terrain area, the overlying rock and soil bodies load on the pillar can be regarded as a vertical uniform load; in the valley terrain area, assuming the single valley slope angle (slope angle) is constant, the effect of the overlying rock and soil layer can be simplified as a linearly distributed load action.

Since the length of the pillar is much larger than the width, the deformation problem of the pillar under the overlying load can be treated as the elastic plane strain problem. Considering that the valley is composed of the slope bottom and two independent slope surfaces, the pillar bearing law is symmetrical, here we only discuss the loading situation of the pillar under a single slope as shown in Figure 2. The calculation method of pillar load under valley terrain is analyzed below.

As shown in Figure 2, the pillar load can be recorded as the weight of the overlying rock and soil bodies in the area corresponding to the pillar (pillar and the central area of the empty area on both sides), and the pillar load P can be calculated by Formula (1).
(1)$$P=\gamma D\left(a+b\right)$$

In the formula: *γ* is the average bulk density of the overlying rock and soil bodies; *D* is the thickness of the overlying rock and soil layer at the center of the pillar (from now on the burial depth of the pillar).

According to the A. H. Wilson pillar bearing theory [31], under the action of the overlying rock load *q*, the stress yield zone will be generated on both sides of the pillar. The central region of the pillar will be the elastic core region (Figure 3). In the yield zone of the pillar, the lateral stress *σ*_3_ gradually increases from the open zone to the inside and reaches the maximum value at the junction of the yield zone and the elastic core zone. The lateral stress at the junction can be regarded as the original rock stress before the open zone mining. size, that is, *σ*_3_ = *γD*.

As shown in Figure 3a, the red real line part is the actual *σ_y_*, and the stress in the yield zone and elastic core zone of the pillar changes nonlinearly; the red dotted line shows part of the assumption *σ_y_* and the stress in the yield zone and elastic core zone of the pillar changes linearly, the total load that the pillar can support under the valley terrain is
(2)$$\begin{array}{l} P_{\max}=\frac{1}{2}d_{1}\sigma_{y_{1}}+\frac{1}{2}d_{2}\sigma_{y_{2}}+\frac{a-d_{1}-d_{2}}{2}\left(\sigma_{y_{1}}+\sigma_{y_{2}}\right)\\ =\frac{a-d_{2}}{2}\sigma_{y_{1}}+\frac{a-d_{1}}{2}\sigma_{y_{2}} \end{array}$$
where *d*_1_ is the width of the yield zone on the left side of the pillar (as shown in Figure 3);

*d*_2_ is the width of the yield zone on the left side of the pillar;

$\sigma_{y_{1}}$ is the vertical stress at the junction between the yield zone on the right side of the pillar and the elastic core zone;

$\sigma_{y_{2}}$ is the vertical stress at the junction between the yield zone on the left side of the pillar and the elastic core zone.

## 3. Pillar Safety Coefficient under Valley Terrain

Equations (1) and (2), respectively, obtain the pillar load calculation formula from the aspect of the pillar bearing the overlying rock-soil mass load (macro) and the aspect of the stress distribution inside the pillar (micro). There should be an equivalent relationship between the two, so the two formulas can be combined to obtain the calculation formula for the limit width of the pillar:(3)$$a_{t}=\frac{d_{2}\sigma_{y_{1}}+d_{1}\sigma_{y_{2}}+2\gamma Db}{\sigma_{y_{1}}+\sigma_{y_{2}}-2\gamma D}$$

As shown in Figure 3b, after the bauxite ore body is mined, the overlying rock and soil layer tends to slide to the goaf under the action of its own weight, Therefore, shear stress *τ_yx_* will be generated at the interface (ore-rock interface) between the pillar and the roof and floor strata. Due to the low strength and poor shear resistance of the bauxite ore body, the pillar will suffer extrusion shear failure under the action of the shear stress at the ore-rock interface. In order to simplify the analysis, it is assumed that the shear failure surface of the bauxite pillar is parallel to the ore bed interface under the action of the overlying rock-soil mass, and the stress unit body on the ore-rock interface should satisfy the stress balance differential equation.
(4)$$\frac{\partial \sigma_{x}}{\partial x}+\frac{\partial \tau_{yx}}{\partial y}=0,\frac{\partial \tau_{yx}}{\partial x}+\frac{\partial \sigma_{y}}{\partial y}=0$$

In addition, the shear stress *τ_yx_* of the ore-rock interface should also comply with the law of sliding friction, that is,
(5)$$\tau_{yx}=-\left(\sigma_{y}\tan\varphi +c\right)$$
where *c* and *φ* are the cohesion and internal friction angle of the rock-soil mass at the ore-rock interface, respectively.

The stress boundary conditions of the pillar should satisfy the following
(6)$$\left\{\begin{array}{l} x=-d_{1},\sigma_{x}=0,\sigma_{y}=0\\ x=0,\sigma_{x}=\gamma D,\sigma_{y}=\sigma_{y}{}_{{}_{1}}\\ x=a-d_{1}-d_{2},\sigma_{x}=\gamma D,\sigma_{y}=\sigma_{y}{}_{{}_{2}}\\ x=a-d_{1},\sigma_{x}=0,\sigma_{y}=0 \end{array}\right.$$

The force in the horizontal direction of the yield zone on both sides of the pillar should be balanced, which
(7)$$\left\{\begin{array}{l} \lambda h{\left[\sigma_{y}\right]}_{x=0}+2\int_{-d_{1}}^{0}\left[\tau_{yx}\right]dx=0\\ \lambda h{\left[\sigma_{y}\right]}_{x=a-d_{1}-d_{2}}+2\int_{a-d_{1}-d_{2}}^{a-d_{1}}\left[\tau_{yx}\right]dx=0 \end{array}\right.$$

In the formula: *λ* is the lateral pressure coefficient of the bauxite ore body, which satisfies *λ* = *μ*/(1 − *μ*), where *μ* is the Poisson’s ratio of the bauxite ore body; *h* is the height of the pillar.

Combined Formulas (4)–(7), the calculation formula for the width of the yield zone of the pillar can be obtained
(8)$$\left\{\begin{array}{l} d_{1}=\frac{\lambda h}{2\tan\varphi }\ln\left(1+\frac{\sigma_{y_{1}}\tan\varphi }{c}\right)\\ d_{2}=\frac{\lambda h}{2\tan\varphi }\ln\left(1+\frac{\sigma_{y_{2}}\tan\varphi }{c}\right) \end{array}\right.$$

According to the research of Zhu et al. [32], it can be assumed that the vertical stress at the junction of the two yield zones of the pillar and the elastic core zone has the following relationship:(9)$$\sigma_{y_{2}}=\frac{\sigma_{y_{1}}}{K}$$

In the formula: *K* is the stress concentration coefficient, which is related to the valley slope angle α, the pillar buried depth *D*, the pillar width a and other parameters. When *K* = 1.0, it means that the pillar does not cause stress concentration due to the valley terrain, that is, the pillar is under the peaceful terrain.

It is assumed that the rock failure obeys the Hoek-Brown strength criterion, namely
(10)$$\sigma_{1}=\sigma_{3}+\sqrt{m\sigma_{m}\sigma_{3}+s\sigma_{m}{}^{2}}$$

Among them, *σ*_1_ and *σ*_3_ are, respectively, the maximum and minimum principal stress when the ore rock is damaged; *σ*_m_ is the uniaxial compressive strength of the complete ore rock; *m* and *s* are dimensionless constants, which are related to the properties of the ore rock and the development of the structural plane, and so on.

Substituting Equations (8)–(10) into Equation (3), the limit width of the pillar under the valley terrain can be obtained by using the structural parameters of the goaf and the occurrence parameters of the ore rock, and so on.
(11)$$a_{t}=\frac{\frac{\mu h\sigma_{y_{1}}}{2\left(1-\mu \right)\tan\varphi }\left[K\ln\left(1+\frac{\sigma_{y_{1}}\tan\varphi }{Kc}\right)+\ln\left(1+\frac{\sigma_{y_{1}}\tan\varphi }{c}\right)\right]+2K\gamma Db}{\left(1+K\right)\sigma_{y_{1}}-2K\gamma D}$$

In the formula,
$$\sigma_{y_{1}}=\gamma D+\sqrt{m\sigma_{m}\gamma D+s\sigma_{m}{}^{2}}$$

The reduction method is used to calculate the safety coefficient of the pillar under the valley terrain.
(12)$$F=\frac{a}{a_{t}}$$

In the formula: *F* is the safety coefficient of the pillar; *a* is the width of the pillar in the actual goaf; at is the limit width of the pillar, and the calculation formula is shown in Formula (11).

Combined with the analysis of Equations (11) and (12) and their parameter meanings, it can be seen that the safety coefficient of the pillar under the valley terrain is affected by the structural parameters of the goaf (the width, span and height of the pillar is *a*, *b* and *h*, respectively), and the property parameters of the ore-rock (Poisson’s ratio *μ*, cohesion of rock-soil mass at the ore-rock interface *c*, internal friction angle *φ*, uniaxial compressive strength *σ*_m_, average bulk density of overlying rock and soil layers *γ*, dimensionless parameters *m* and *s*), valley terrain parameters (valley slope angle *α*, pillar burial depth *D*, stress concentration coefficient *K*).

## 4. Parameter Influence Analysis

As far as the goaf of a sedimentary bauxite mine in a specific area is concerned, the composition and properties of the surrounding rock and the ore rock are often the same. This section mainly studies the influence of two types of parameters, the structural parameters of the goaf and the valley terrain parameters, on the stability of the pillar under the valley terrain [33,34,35]. Considering that the spatial relationship between the direction of the goaf and the valley will change the values of the two types of parameters, this section will also analyze and discuss the influence of the spatial relationship between the two on the stability of the pillar.

Taking a sedimentary bauxite mining area underlying a typical valley terrain in Shanxi as the research object, the mine adopts the strip room-pillar method for mining. According to the measured data of the mine, the structural parameters of the goaf and the physical and mechanical parameters of the rock and soil are shown in Table 1. For comparative analysis, except for the independent variables, it is assumed that the strike of the goaf is parallel to the strike of the valley, the pillar width a is 4.5 m, the goaf span b is 7.5 m, the pillar height h is 4.5 m, and the pillar burial depth *D* is 70 m, the stress concentration coefficient *K* is taken as 1.4.

### 4.1. Influence Analysis of Structural Parameters of the Goaf

Figure 4 and Figure 5 show the variation curves of the pillar safety coefficient *F* with the goaf span *b* and the pillar height *h* under the conditions of different stress concentration coefficients *K* calculated by Formula (12).

**Table 1 ijerph-19-10991-t001:** Relevant parameters of rock and soil at the ore-rock interface.

Parameters	Cohesion *c*/MPa	Internal Friction Angle *φ*/°	Test Weight *γ*/kN·m^−3^	Poisson’s Ratio *μ*	Constant m	Constant s	Uniaxial Compressive Strength *σ*_m_/MPa
Value	1.29	31.7	22.5	0.24	6.7	0.3	26.86

It can be seen from Figure 4 that under the conditions of stress concentration coefficient *K* is 1.4 and 1.8, the safety coefficient *F* of the pillar under the valley terrain decreases in a linear relationship with the increase of the span *b* of the goaf, this is consistent with the change rule of the pillar safety coefficient under peaceful terrain (*K* = 1.0). The reason is that the more extensive *b* is, the greater the ore recovery rate is, the greater the pillar load is, and the easier it is to induce the pillar instability.

In addition, it can also be seen in Figure 4 that when the value of *K* increases (*K* = 1.0, 1.4, 1.8), as b increases from 4.5 m to 9.0 m, *F* decreases from 1.32, 1.37, 1.42 to 1.20, respectively, 1.23, and 1.26, and the reduction rates were 8.5%, 10.2%, and 11.6%, respectively. The decrease of *F* increases with the increase of *K*, indicating that the stress concentration caused by the valley terrain makes the pillar more likely to induce instability when the span of the goaf increases, and the stress concentration is more important (the larger the *K*), the pillar is more prone to instability. Therefore, for the overlying rock and soil of the ore body with significant valley terrain characteristics, the span of the goaf should be appropriately reduced to ensure the safety and stability of the pillar [36].

In Figure 5, under the given condition of stress concentration coefficient *K*, the safety coefficient *F* of the pillar under the valley terrain changes with the height *h* of the pillar, and the law is also consistent. *F* decreases with the increase of *h*, and the decreasing rate decreases. The two are approximately in a negative exponential relationship. This shows that the law of pillar stability affected by pillar height under valley terrain is the same as that under mild terrain conditions. Due to the effect of pillar size, when the pillar width is constant, the greater the height of the pillar, the smaller the aspect ratio of the pillar, the weaker the bearing capacity of the pillar, and the lower the safety coefficient of the pillar [37,38,39].

It can also be found in Figure 5 that under different *K* values (*K* = 1.0, 1.4, 1.8), when *h* increases from 3.0 m to 5.0 m, *F* decreases from 1.44, 1.47, 1.51 to 1.13, 1.16,1.20, respectively, and the reduction rates are 34.7%, 33.5%, and 32.6%, respectively. The decreasing range of *F* decreases with the increase of *K*, indicating that the stress concentration effect caused by the valley terrain can resist the pillar instability caused by the pillar size effect to a certain extent, that is, as h increases, the stress concentration effect is more significant (the larger the *K*), the less likely the pillar will be unstable. Therefore, in order to improve the mining rate of the ore body and at the same time ensure the safety and stability of the pillar under the valley terrain, taking into account the effect of the valley terrain, the method of appropriately increasing the pillar height can be adopted.

But it is worth noting that when *K* increases from 1.0 to 1.8, the change in the reduction range of *F* is only 2.1%, which is 6.1% of the reduction range. It can be seen that the resistance of stress concentration to the pillar instability caused by the increase of h is relatively tiny.

### 4.2. Influence Analysis of Valley Terrain Parameters

Figure 6 shows the change curve of the pillar safety coefficient *F* under the valley terrain under different pillar burial depths *D* when the stress concentration coefficient *K* is taken as 1.0, 1.4 and 1.8, respectively. It can be seen from Figure 6 that under different *K* values, both *F* and *D* are negatively correlated, and decrease as *D* increases, and the rate of decrease gradually slows down. Since local ore bodies can be regarded as buried in the same horizontal position, the thickness of rock and soil mass in the valley area changes with the spatial position, so the safety coefficient of the pillar under the valley terrain also changes accordingly. The safety coefficient of the pillar located in the upper section of the valley is rather small, and the safety coefficient of the pillar located in the left side of the valley is relatively large.

It can also be seen from Figure 6 that when *K* is 1.0, 1.4 and 1.8, respectively, when *D* increases from 30 m to 120 m, *F* decreases from 1.48, 1.56, 1.64 to 1.07, 1.08, and 1.10, respectively, with a reduction range of 27.6%, 30.5%, 32.8%. Compared with the peaceful terrain condition (*K* = 1.0), as *D* increases, the decreasing range of *F* increases gradually, indicating that the influence of valley terrain on the stability of the pillar gradually decreases as the burial depth of the pillar increases. Therefore, in practical engineering, for bauxite and other ore bodies with shallow burial depth, the influence of *D* should be paid attention to when studying the stability of pillar in the right side of the valley, especially at the bottom of the valley.

From the analysis of the parameters *b*, *h*, and *D* above, it can be known that the stress concentration coefficient *K* has an influence on the stability of the pillar. According to the data calculated by Formula (12), the variation curve of the pillar safety coefficient.

F with the stress concentration coefficient *K* under the conditions of different goaf spans b is shown in Figure 7. It can be seen from Figure 7 that under different values of *b*, *F* is in an increasing relationship with *K*. This is because in the data calculation, for comparative analysis, it is assumed that the burial depth *D* of the pillar remains unchanged, and the vertical stress at the junction between the yield zone and the elastic core zone of the pillar on the right side remains unchanged, Therefore, the rise of *K* is essentially equivalent to the increase of the valley slope angle *α*, which leads to a decrease of $\sigma_{y_{2}}$, which in turn leads to a corresponding decrease in the pillar load (that is, a decrease in the effective bearing capacity of the pillar), resulting in an growth in *F*.

To further quantitatively study the influence of the valley slope angle α on the stability of the pillar, considering that the slope of the overlying vertical stress curve of the lower pillar of the valley loess is consistent with the slope of the valley slope, the following formula can be used to calculate the stress concentration coefficient:(13)$$K=\frac{\sigma_{y}{}_{{}_{1}}}{\sigma_{y}{}_{{}_{1}}-a\tan\alpha }$$

According to Formula (13), the stress concentration coefficient *K* value under the conditions of different pillar widths and different valley slope angle *α* is obtained as shown in Figure 8. When the burial depth *D* of the pillar and the span *b* of the goaf are the same, with the increase of *α*, *K* gradually increases, and the rate of increase of *K* increases exponentially, indicating that the larger the *α*, the greater the stress concentration result of the pillar caused by the valley terrain, and the result is particularly significant when the *α* is large (*α* > 40°). It can also be seen from Figure that when the pillar width an increase, the rate at which *K* increases with the increase of *α* also increases gradually. Since *K* in Formula (13) is the ratio of ultimate stress on both sides of the pillar, this phenomenon shows that when an increases, *K* will be easier to reach the same value, which is consistent with the engineering practice.

Further, under pillar width a, the change curve of the pillar safety coefficient *F* with the valley slope angle *α* under the valley terrain is shown in Figure 9. It can be seen from the figure that it is similar to the change curve of *K* with *α*. Under the same conditions, as *α* increases, *F* gradually increases, and the rate of growth of *F* increases exponentially, especially when *α* is large, the change of *F* is particularly severe. This is because under a specific pillar ultimate strength, the more significant the *α*, the more prominent the stress concentration in the pillar, and the corresponding reduction of the pillar load, which leads to the larger *F*; according to Figure 7, the more significant the *α* is, the more significant the stress concentration of the pillar is, and the more severe the change of *F* is. Therefore, when studying the stability of pillar under valley terrain, we should pay attention to the effect of the valley slope angle α, in particular, we should pay close attention to the effect of the valley slope angle α on the pillar when the valley slope angle α is large.

### 4.3. Influence Analysis of the Goaf Strike

Compared with the pillar under peaceful terrain, when the ore bed is located in the valley terrain area, the study of pillar stability has its particularity [40,41,42]. Next, the influence of the spatial relationship of goaf and valley on the stability of the pillar is studied.

There are three kinds of relationships between the strike of the goaf (that is, the strike of the pillar) and the strike of the valley, which are parallel, oblique, and vertical. In order to uniformly describe the spatial relationship between the strike of the goaf and the valley in the three cases, a schematic diagram of the spatial relationship model between the strike of the goaf and the valley as shown in Figure 10 is established. In the figure, *α′* is the real slope angle of the valley at the corresponding position of the goaf strike; *β* is the angle between the goaf strike and the valley strike, and *α* is the valley slope angle.

According to the transformation law of solid geometry, the three should satisfy the following relationship:(14)tanα′=sinβ•tanα

According to Formula (14), when *β* = 0°, the goaf strike is parallel to the valley strike. At this time, the real valley slope angle *α′* at the corresponding position of the goaf strike is the same as the valley slope angle *α*, and the previous analysis method can be directly used to study the stability of the pillar; when 0° < *β* < 90°, the strike of the goaf is parallel to the strike of the valley. At this time, we only need to replace *α* with *α′* in the previous analysis method to obtain the safety coefficient of the pillar in the case of oblique intersection.; when *β* = 90°, the strike of the goaf is perpendicular to the strike of the valley. Considering that the strike length of the strip pillar in the goaf is much larger than the width of its layout, it can be assumed that the pillar is under peaceful terrain for comparative analysis.

Figure 11 shows the variation curve of stress concentration coefficient *K* under different spatial relationship conditions of goaf and valley. In the same way as *K* changes with *α*, when the value of α is given, as the angle *β* between the strike of the goaf and the valley increases, *K* gradually increases (Figure 11), because *β* affects the real valley slope angle *α′*. The process of growing *β* from 0° to 90° is also the process of growing α′ from 0° to *α*. It can also be seen in Figure 11 that when *β* is small (*β* < 50°), *K* and *β* are approximately linearly related. when *β* is large (*β* > 50°), *K* and *β* are approximately parabolic, the increase rate of *K* gradually slows down. This shows that compared with when *β* is enormous, when *β* is small, it has a more significant influence on the stress concentration of pillar, and the effect is gradually weakened when *β* is large.

To study the effect of different spatial relationship between goaf and valley on pillar stability, under the condition of different pillar widths *a*, the change curve of the pillar safety coefficient *F* with the angle *β* between the goaf strike and the valley strike under the valley terrain is shown in Figure 12. It can be seen from the figure that, similar to the variation law of *K* with *β*, under a certain condition of *α*, F gradually increases with the increase of *β*. As *β* increases, *F* first increases approximately linearly, and then increases. This shows that at the same valley position, the oblique method of goaf strike and valley strike is used to mine the ore body, which can improve the stability of the pillar, especially when the *β* is small, the improvement effect is more pronounced.

It can also be seen in Figure 12 that as *β* increases, when *α* is 30°, 45° and 60°, the magnitude of the increase in *F* also increases, indicating that the steeper the slope of the valley, the greater the influence of the goaf strike on the stability of the pillar under the valley terrain. Therefore, the spatial relationship between the goaf, and the valley should be taken into consideration when studying the stability of the pillar under the valley terrain.

To sum up, in the research and calculation of the stability of the underlying sedimentary bauxite pillars in the actual valley terrain, not only the calculation based on the structural parameters of the goaf, but also the influence of the valley terrain parameters should be taken into consideration, in addition, the effect of the goaf strike should be analyzed.

## 5. Engineering Verification

A sedimentary bauxite mining area in Shanxi is a typical valley terrain. The gully in the area is dense and narrow, and the shape is a “V” shape. It is distributed alternately with loess beams, replat, and walls, and micro-topographic landscapes such as cliffs, loess residual pillars, and sink holes are familiar (Figure 1). In general, the terrain of the mining area is high in the east and low in the west. The top of the back side beam in the east of the site is the highest point of the terrain, with an elevation of +1084 m. The bottom of the valley of Guandi Cliff in the west of the area is the lowest point, with an elevation of +935 m and a maximum relative height difference of 149 m. The mine is mainly mined by the strip room-pillar method [43,44,45]. After years of mining, many ore pillars and empty areas have been left under the valley terrain.

Select the series of pillars on the 6# exploration line (Figure 13) as the object for analysis. According to the mine site survey results, the mine pillar width *a* is 4.5 m, the goaf span *b* is 7.5 m, the pillar height *h* is 4.5 m, the goaf strike is approximately parallel to the valley strike, the elevation of the highest point on the exploration line is +1069 m, the elevation of the lowest point is +954 m, and the valley slope angle *α* is 23~49°. If the average mining depth is 80 m and the maximum valley slope angle is 49°, and the limit width of the pillar is calculated according to Formula (11), the safety coefficient of the pillar can be further obtained as 1.21, and the pillar is in a stable state.

If considering the influence of valley terrain, the minimum thickness of the overlying rock and soil bodies are less than 20 m, and the safety coefficient of the pillar can be calculated as 1.68. The safety margin is too large (the safety coefficient is generally taken as 1.20~1.40), which will cause waste of bauxite ore resources. In addition, the thickness of the overlying rock and soil bodies in many pillars can be as high as 130 m. The stress concentration coefficient *K* is 1.41, and the pillar safety coefficient is 1.05. It can be seen that the stress concentration of the pillar is relatively significant, and the pillar safety coefficient is close to the critical value of the stable state. 1.00. Pillar instability did not occur in the short term after forming the pillar and goaf in this area. However, after the goaf was exposed, the pillar was affected by weathering and the stress concentration of the overlying load. Local collapse began to occur on the right side of the pillar (Figure 14), until the pillar became unstable, causing the collapse of goaf (Figure 15).

## 6. Conclusions

In this paper, aiming at the vertical and horizontal characteristics of the surface valleys in the sedimentary bauxite mining area, the mechanical model of the pillar under the valley terrain is constructed, the calculation formula of the safety coefficient of the pillar under the valley terrain is established, the law that the pillar stability of a typical sedimentary bauxite mine in Shanxi is affected by the surface valley terrain is analyzed. The main conclusions are as follows:
(1)The overlying valley terrain in the sedimentary bauxite mining area causes stress concentration in the pillar, reduces the effective bearing capacity, and reduces the stability of the pillar. The influence of the surface valley terrain should be considered when studying the stability of the pillar;(2)The analysis of the influence of the structural parameters of the goaf shows that, due to the stress concentration caused by the valley terrain, the mine pillar is more likely to induce instability when the span *b* of the goaf rise; the more significant the stress concentration effect is, the more likely the instability phenomenon will occur. The pillar is more prone to instability when the pillar height *h* increases, but the stress concentration effect is less resistant to the pillar instability caused by the increase of *h*;(3)The analysis of the influence of the valley terrain parameters shows that with the increase of the burial depth *D* of the pillar, the stability of the pillar is weakened, and the influence of the valley terrain gradually decreases; The more significant the valley slope angle *α*, the greater the stress concentration effect of the pillar, the more significant the safety coefficient *F* of the pillar under the same burial depth, and the greater the influence of *α* on *F*;(4)The analysis of the influence of the goaf strike shows that, compared with when the angle between the goaf strike and the valley strike is *β* > 50°, when *β* < 50°, the result of *β* on the stress concentration and safety coefficient of the pillar is greater, and the result is gradually weakened when *β* is larger. In addition, the steeper the slope of the valley, the more significant the influence of the goaf strike on the stability of the pillar under the valley terrain;(5)The safety coefficient analysis results show that the safety coefficient of the pillar at the bottom of the valley on the 6# exploration line of a bauxite mine in Shanxi has too much margin, while the safety coefficient of the pillar at the top of the valley is close to the critical value, and the pillar is on the edge of instability, the field investigation results verify the reliability of the quantitative assessment method of pillar stability in this paper.

## Figures and Tables

**Figure 1 ijerph-19-10991-f001:**
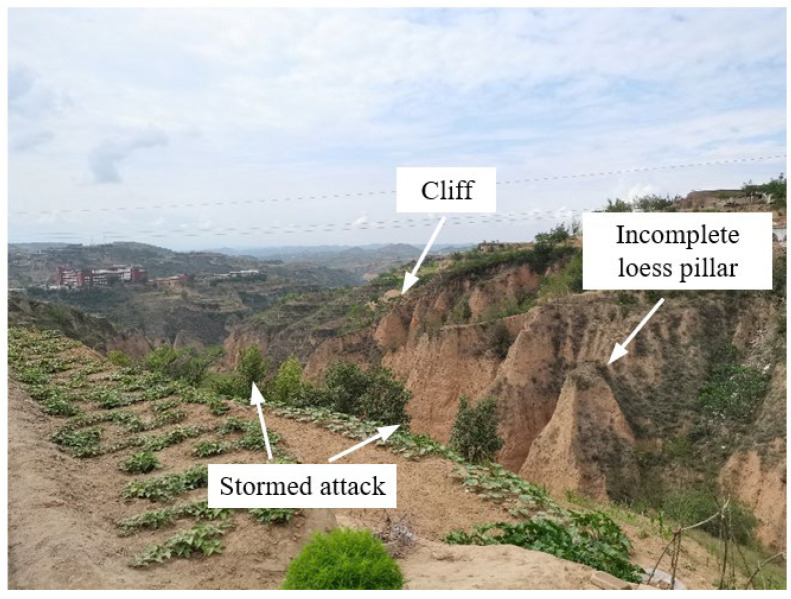
The actual valley terrain in a sedimentary bauxite area in Shanxi.

**Figure 2 ijerph-19-10991-f002:**
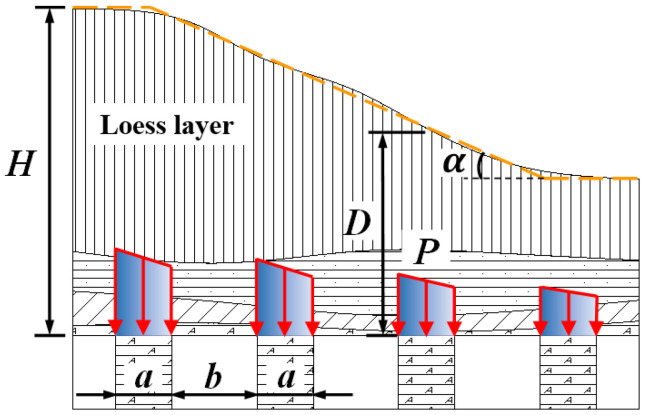
Schematic diagram of goaf space structure and pillar load.

**Figure 3 ijerph-19-10991-f003:**
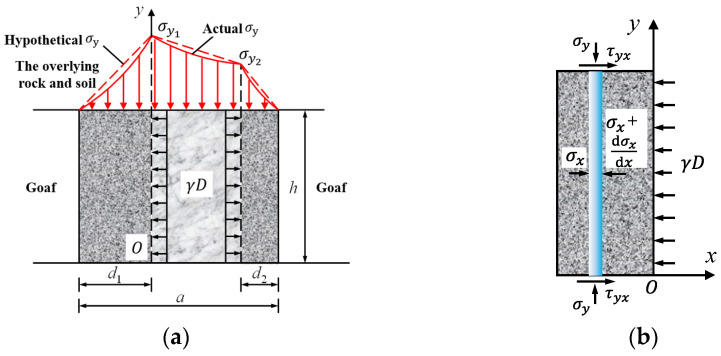
Mechanical model of pillar under valley terrain: (**a**) schematic diagram of mechanical analysis of pillar; and (**b**) schematic diagram of mechanical analysis of yield zone (the left yield zone).

**Figure 4 ijerph-19-10991-f004:**
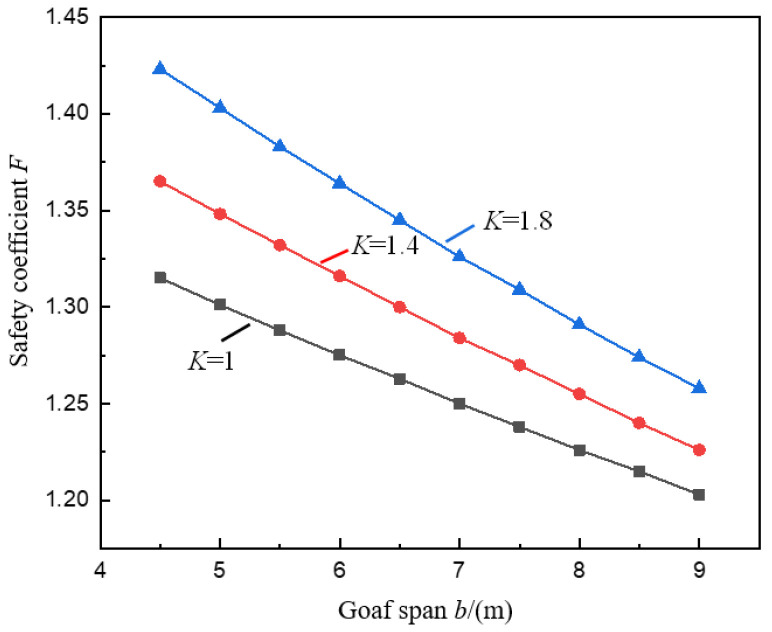
Variation curve of pillar safety coefficient *F* with goaf span *b*.

**Figure 5 ijerph-19-10991-f005:**
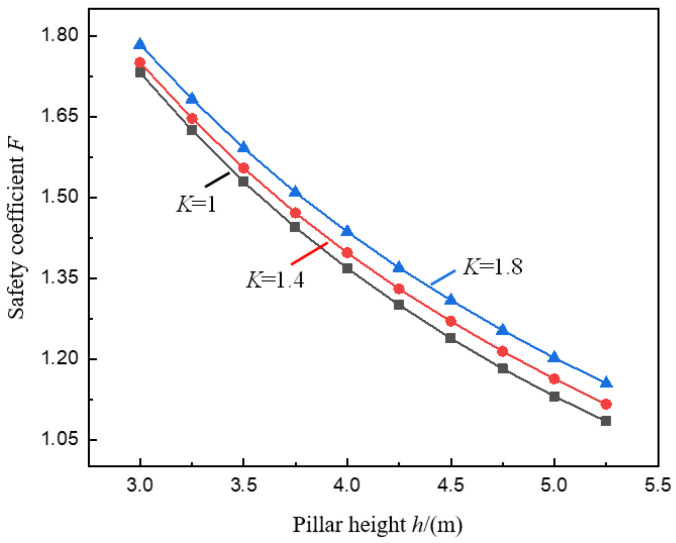
Variation curve of pillar safety coefficient *F* with pillar height *h*.

**Figure 6 ijerph-19-10991-f006:**
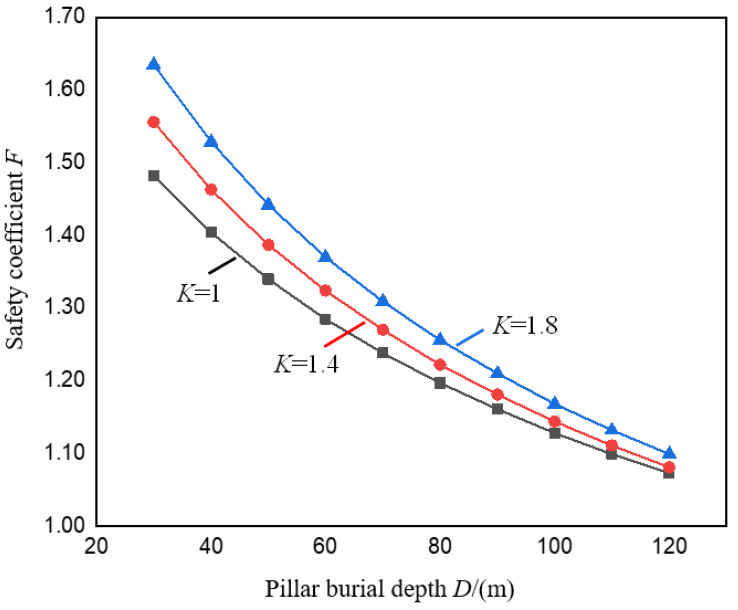
Variation curve of pillar safety coefficient *F* with pillar burial depth *D*.

**Figure 7 ijerph-19-10991-f007:**
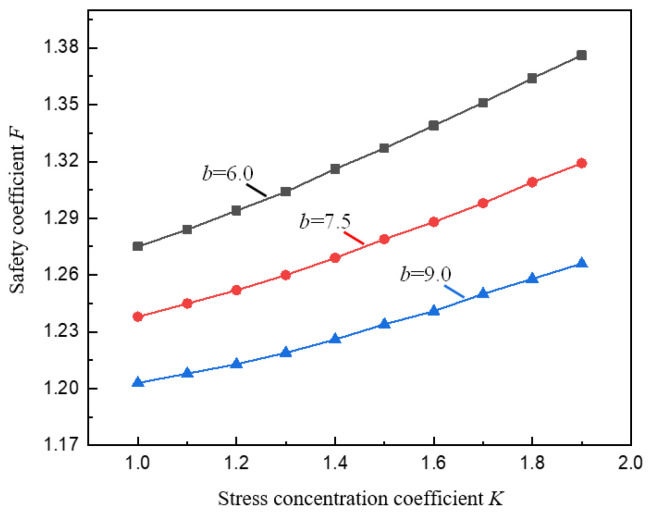
Variation curve of pillar safety coefficient *F* with stress concentration coefficient *K*.

**Figure 8 ijerph-19-10991-f008:**
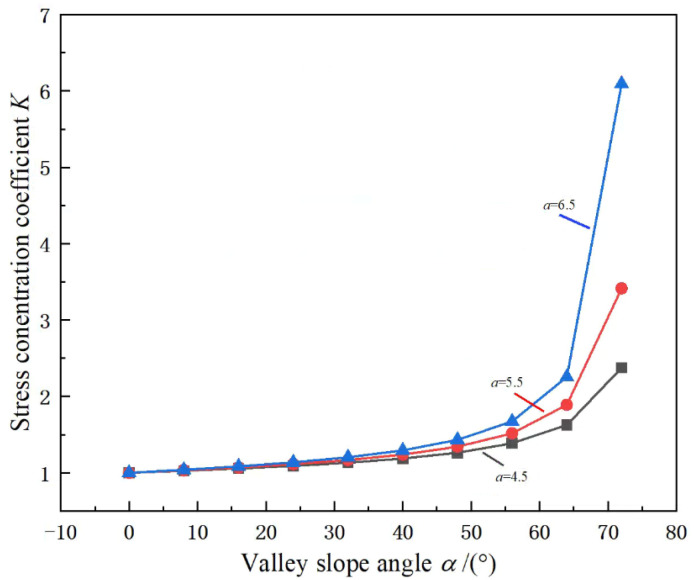
Variation curve of stress concentration coefficient *K* with valley slope angle *α*.

**Figure 9 ijerph-19-10991-f009:**
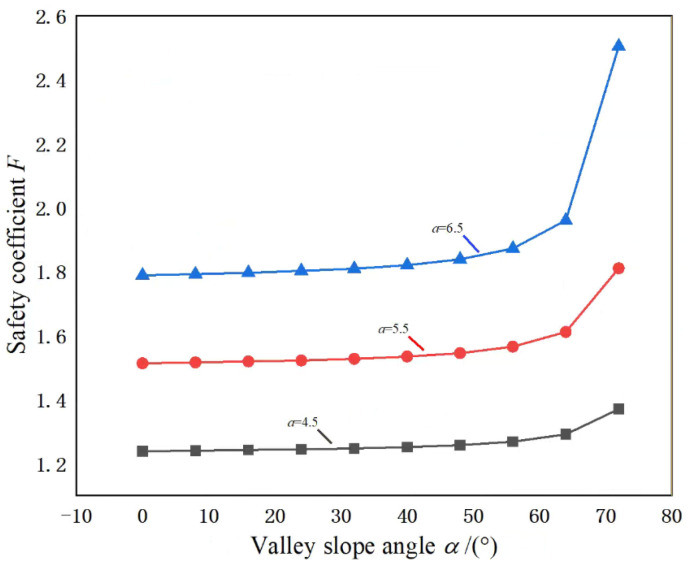
Variation curve of pillar safety coefficient *F* with valley slope angle *α*.

**Figure 10 ijerph-19-10991-f010:**
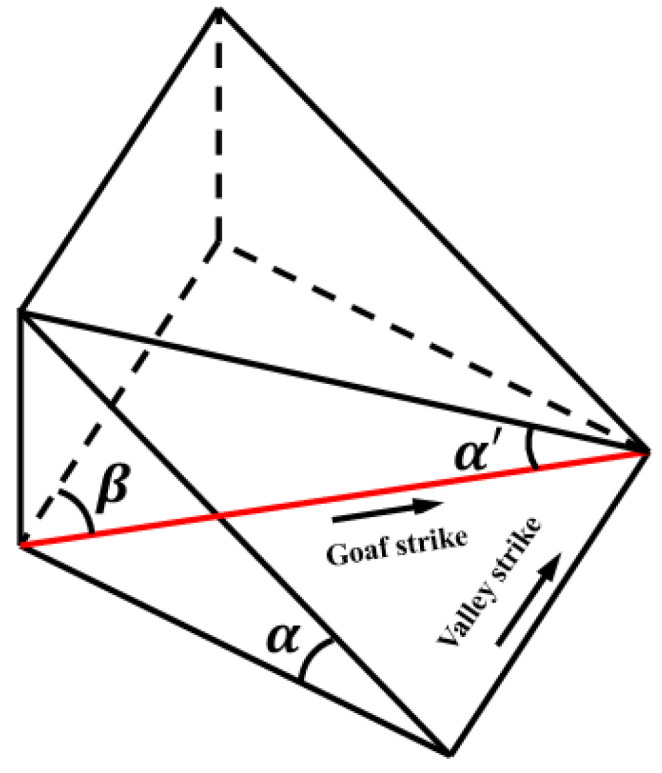
Spatial relationship between goaf strike and valley.

**Figure 11 ijerph-19-10991-f011:**
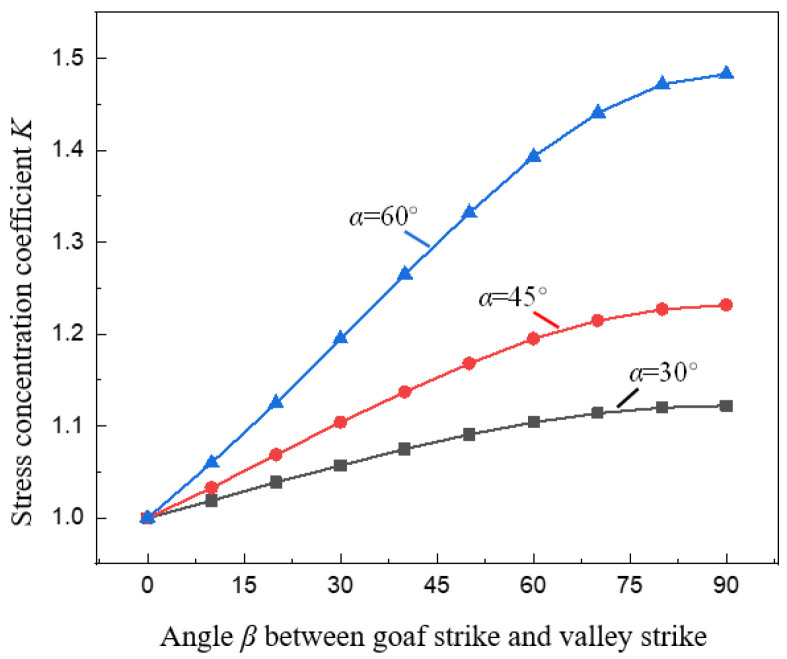
Relationship between goaf strike and stress concentration coefficient *K*.

**Figure 12 ijerph-19-10991-f012:**
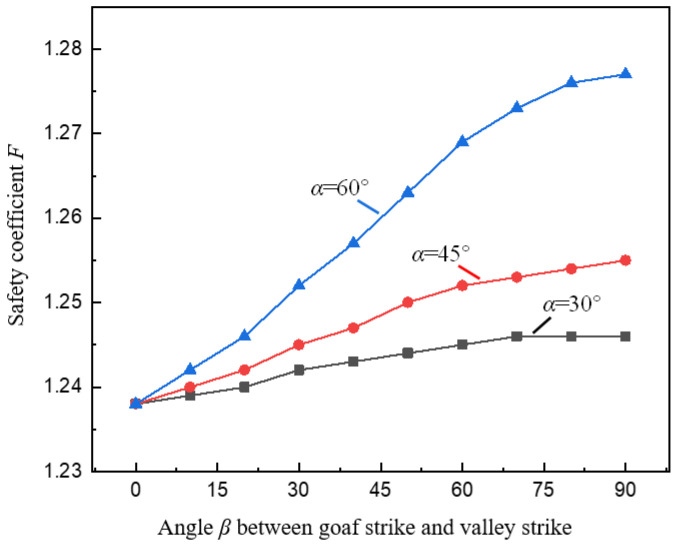
Relationship between goaf strike and stress concentration coefficient *F*.

**Figure 13 ijerph-19-10991-f013:**
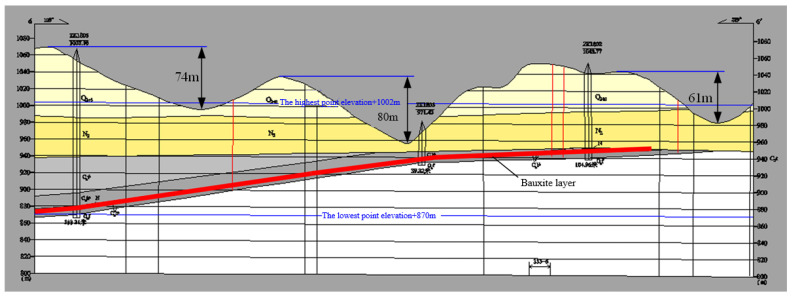
Sectional view of 6# exploration line of a sedimentary bauxite mine in Shanxi.

**Figure 14 ijerph-19-10991-f014:**
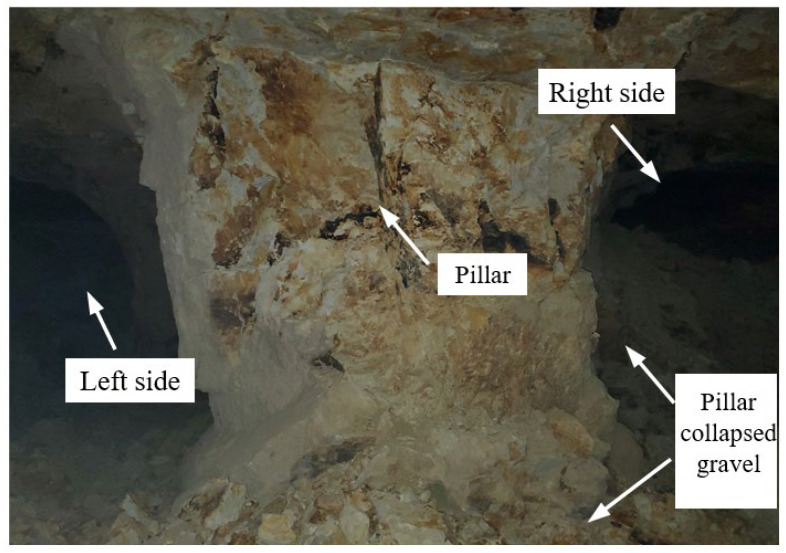
Partial collapse of pillar.

**Figure 15 ijerph-19-10991-f015:**
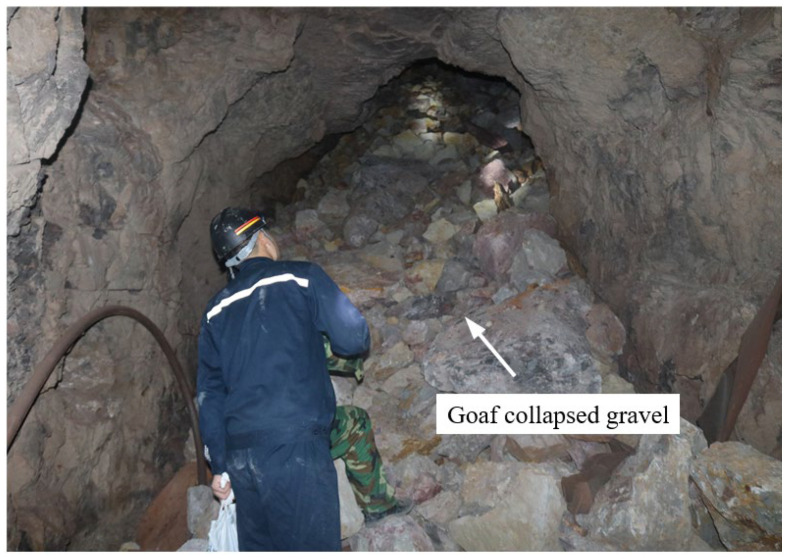
Collapse of goaf.

## Data Availability

Not applicable.

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
