# Peer review of "Study on Safety Coefficient of Sedimentary Bauxite Strip Pillar under Valley Terrain"

_ijerph, 2022, doi:10.3390/ijerph191710991_

Round 1
Reviewer 1 Report
The underlying goaf under the valley landform is a common phenomenon in the mining of sedimentary bauxite in Shanxi, and many problems such as stress concentration and shear-slip failure often occur in pillar under the valley terrain, which seriously threaten the safety and stability of the goaf. Therefore, it is of great practical significance to grasp the influence law of valley terrain on the stability of the underlying sedimentary bauxite strip pillars to ensure the safe mining of such mines. In this paper, taking a sedimentary bauxite mining area under a typical valley terrain in Shanxi as an example, a pillar mechanics model is constructed based on the force characteristics of the valley terrain, and a calculation method of the pillar safety factor under the influence of the structural parameters of the goaf and the valley terrain is proposed. Finally, combined with engineering examples, the safety factor analysis was carried out to verify the reliability of the method for quantitative assessment of pillar stability. The research ideas of the paper are correct, the arguments are sufficient, and the diagrams are clear.
I agree to publish the paper, but revised the related questions as following:
1: Supplement the basic geological conditions of the overlying rock and soil layers in the goaf of sedimentary bauxite.
2: The analysis of influencing factors in Section 4, the paper does not explain the influence of hydraulic infiltration on the safety factor of strip pillars, please add.
3: In the paper, please mark the descriptions in Fig.1, Fig.2, Fig.3, Fig.10, Fig.13, Fig.14, and Fig.15 in English.
4: In Fig.14, both sides of the mine pillar should be goaf, not the uphill side and the downhill side.
5: Some of the black numbers in Figure 13 are difficult to distinguish, please replace them with clearer figures.
6: In the text, 176 rows of lateral stress "σ3" and 246 rows of maximum principal stress "σ1" should be changed to subscript format.
7: "Fig.x" and "Figure.x" appear many times in the text, please unify the format specification.
8: Please keep the format consistent for the literature citation problems in the introduction, such as "Zhao Kang et al [13] …", "Luo Binyu et al. [14] ..." and "Li et al. [16] …","Wang et al. [17] …".
9: Unify the formula format in the text, as well as the symbols and subscripts that appear. For example, "d1" in line 190 should be subscripted.
Reviewer 2 Report
The manuscript describes the stress analysis and the safety level evaluation of a sedimentary bauxite strip pillar under valley terrain, as indicated in the title.
The authors describe the geometric hypotheses and appropriately explain the simplifications adopted.
The mechanical analyzes are quite simplified and this allows to describe the effects of the variation of the parameters in a direct, analytical way.
The authors show a set of graphics that resume their finding as a function of the main parameters that characterize the very simplified geometry of their model, as well as the stress concentration coefficient K.
The conclusions are consistently demonstrated and they are not surprising.
A part that does not seem completely clear, from the point of view of the notation, concerns the subscript indices of the variables that indicate the stress components are not clear and univocal (pag 4). For example, compare the text of section 2 with what is indicated in figure 3.
The present reviewer, however, feels unable to grasp the innovative and general aspects of this presentation. In general, the manuscript does not show that there is a research with a clear design in terms of scientific or technological enhancement. It seems more something like a technical report.
Perhaps this is due to my limited expertise in the field of geo-technical and mining engineering, so I will limit my observation strictly to aspects that pertains to computational solid mechanics or dynamics of structures, and in both these fields the manuscript does not seem to propose a significantly innovative approach or results for the international technical-scientific community.
However, it is possible that my point of view does not fully understand other aspects that are of greater interest in the field of geo-technical or mining engineering research.
Round 2
Reviewer 2 Report
The manuscript has been improved and is acceptable for publication.